# A Plug-In Curriculum Scheduler for Improved Deformable Medical Image Registration

## Abstract

Deformable image registration is a crucial task in medical image analysis, and its complexity has spurred significant research and ongoing progress. Much of the work in this area has concentrated on achieving incremental performance gains by adjusting network architectures or introducing new loss functions. However, these modifications are often tailored to specific tasks or datasets, which limits their general applicability. To address this limitation, we propose an innovative solution: a plug-in curriculum scheduler that can be seamlessly integrated into existing methods without changing their core architecture. Our scheduler, inspired by curriculum learning, progressively increases task difficulty to enhance performance, incorporating sample difficulty and matching accuracy as key criteria. Sample difficulty is assessed at voxel and volume levels, using Variance of Gradients for voxel complexity and Gaussian blurring for volume evaluation, while matching accuracy involves gradually increasing supervision for improved alignment and accuracy. We empirically demonstrate that this scheduler achieves superior accuracy and visual quality in various tasks and datasets.

## 1 Introduction

Deformable medical image registration is vital for medical image analysis because it allows precise alignment of images from various times or modalities. This accuracy is key for identifying changes, planning treatments, and combining data from different imaging sources. The field's complexity and its significant role in diagnosis, treatment, and personalized care have led to extensive research and ongoing advancements over the years. However, because of the difficulties in accurately representing deformation fields, research in deformable medical image registration has primarily focused on incremental performance improvements. These improvements often involve minor changes to network architectures, the integration of hierarchical or iterative processes, or the addition of loss functions. In most cases, these methods follow a common structure: a network inputs two images and generates a deformation field to align them. The loss function generally consists of an image similarity measure combined with a regularization term for the deformation field.

Recent studies continue to follow this similar flow. For example, H-ViT Ghahremani et al. (2024) introduces a top-down approach for estimating deformation fields by capturing multi-scale short- and long-range flow features, utilizing dual self- and cross-attention to enhance low-level features with high-level representations. CorrMLP Meng et al. (2024) presents the first correlation-aware MLP-based network for deformable medical image registration, improving efficiency and capturing long-range dependencies at full resolution without using self-attention. IIRP-Net Ma et al. (2024) develops a pyramid registration network that integrates a feature extractor with residual flow estimators to improve the generalization of feature extraction and registration. Despite the difference in architectures used to generate the registration field, all three methods share a common framework; a network for estimating deformation fields, with losses that include an image similarity measure and a regularization term. In addition to these recent methods, a summary of the taxonomy of representative approaches is provided in Table 1.

While modifying model architectures can lead to performance improvements, these approaches are often tailored to specific tasks or datasets, limiting their broader applicability. To overcome this

| Method | CNN | U-Net | Encoder-Decoder | Transformer | MLP |
|---|---|---|---|---|---|
| plain | | VoxelMorph, DALT, SVF-Net, AVSM, LKU-Net, AUM-Net, MIFR, MIDIR | ADRIR | TransMorph, ViT-V-Net | |
| dual / inverse | CycleMorph | ICNet, SYMNet | | DTN | |
| iterative / recursive | IIRP-Net | VTN, RCN, VR-Net | | | |
| hierarchical / pyramid / coarse-to-fine | SDHNet, DLIR | Dual-PRNet | LapIRN, PMDR, Im2grid, NICE-Net | H-ViT, PIViT, Deformer, C2FViT, ModeT | CorrMLP |
| patch-wise | DIRNet | | Quicksilver | TM-DCA, XMorpher | |

Table 1: Taxonomy of design choices for deep learning based medical image registration methods.

limitation, we propose a novel approach that maintains the integrity of existing model structures. Rather than altering the architecture, we introduce a plug-in curriculum scheduler that integrates seamlessly with current methods. Our scheduler is inspired by curriculum learning, a training strategy that involves progressively increasing the difficulty of tasks. This approach has proven effective in various fields, which demonstrated how gradually evolving tasks and network architectures can benefit language processing models. We extend this concept to medical image registration, showing that it can significantly enhance performance on complex tasks.

Our approach incorporates two key criteria to guide the network's learning: sample difficulty and matching accuracy. Sample difficulty involves gradually selecting more challenging training examples, while matching accuracy focuses on ensuring precise alignment by strengthening supervision over time. Sample difficulty is assessed at two levels: voxel and volume. For voxel difficulty within an MRI volume, we use the Variance of Gradients (VoG) to prioritize more complex voxels for better alignment as training progresses. Evaluating difficulty across MRI volumes is more challenging, so we simplify the samples using Gaussian blurring, starting with highly blurred images and gradually shifting to sharper ones. In terms of matching accuracy, we progressively increase the strictness of supervision. This approach allows for more flexibility in the early stages, focusing on broader patterns, and then demands closer adherence to ground truths as training advances, ultimately improving accuracy.

Unlike most curriculum learning methods in medical image analysis, our approach offers an automated curriculum learning solution for medical image registration that does not require expert knowledge or prior experience. This scheduler enhances adaptability and flexibility, enabling improved performance across various tasks and datasets. By focusing on a plug-in curriculum scheduler rather than architectural changes, we aim to develop more robust and widely applicable registration techniques. In summary, our main contributions are as follows.

- We propose a novel approach that introduces a plug-in curriculum scheduler, allowing for seamless integration with existing model structures without modifying their architecture.

- In contrast to most curriculum learning methods in medical image analysis, our approach provides an automated curriculum learning solution for medical image registration that operates without the need for expert knowledge or prior experience.

- This scheduler improves adaptability and flexibility for better performance across diverse tasks and datasets, focusing on robust registration techniques without requiring architectural changes.

## 2 RELATED WORKS

### 2.1 DEFORMABLE MEDICAL IMAGE REGISTRATION

**Traditional Algorithms.** Traditional algorithms for image registration include popular models such as elastic registration Kybic (2001); Shen & Davatzikos (2002); Zhang et al. (2013); Heinrich et al. (2015), b-spline registration Jiang & Shackleford (2015); Sorzano et al. (2005); Zufeng et al. (2016); Modat et al. (2010), and viscous fluid-flow registration Bro-Nielsen & Gramkow (1996); D'agostino et al. (2003). These methods typically involve numerical optimization steps that iteratively adjust a predefined transformation space to find the optimal solution based on explainable metrics. Optical

flow models Brox & Malik (2010); Chen et al. (2013); Ranjan & Black (2017) treat the moving and fixed images as continuous time samples of a sequence. To address challenges with large deformations and reverse fields, enhancements have been made through algorithms such as the Demons algorithm Thirion (1998) and diffeomorphism techniques Ashburner (2007); Avants et al. (2008); Dalca et al. (2018); Janssens et al. (2011); Krebs et al. (2019); Zhang (2018); Beg et al. (2005). These approaches provide smooth, continuous, and invertible velocity fields while preserving the topological structure of images.

**Deep Learning Image Registration (DLIR) Algorithms.** As shown in Tab. 1, DLIR algorithms can be categorized into five types based on their network architectures. Over the past decade, CNNs have been a primary focus of research in medical image registration, including approaches like CycleMorph Kim et al. (2021), IIRP-Net Ma et al. (2024), SDHNet Zhou et al. (2023a), DLIR de Vos et al. (2019), and DIRNet de Vos et al. (2017). U-Net architectures, along with encoder-decoder models, have become popular choices in DLIR due to their efficiency in capturing hierarchical features at multiple resolutions, which is crucial for accurately modeling complex image transformations. Notable examples include VoxelMorph Balakrishnan et al. (2019; 2018); Dalca et al. (2018), DALT Zhao et al. (2019a), SVF-Net Rohé et al. (2017), AVSM Shen et al. (2019), LKU-Net Jia et al. (2022), AUM-Net Xu et al. (2020), MIFR Shin & Lee (2023), MIDIR Qiu et al. (2021), IC-Net Zhang (2018), SYMNet Mok & Chung (2020b), VTN Zhao et al. (2020), RCN Zhao et al. (2019b), VR-Net Jia et al. (2021), and Dual-PRNet Kang et al. (2022) for U-Net, as well as ADRIR Hu et al. (2018), LapIRN Mok & Chung (2020a), PMDR Krebs et al. (2019), Im2grid Liu et al. (2022), NICE-Net Meng et al. (2022), and Quicksilver Yang et al. (2017) for encoder-decoder structures. With the advancement of Vision Transformers (ViT), transformers have also been applied in DLIR, with examples including TransMorph Chen et al. (2022a), ViT-V-Net Chen et al. (2021), DTN Zhang et al. (2021), H-ViT Ghahremani et al. (2024), PIViT Ma et al. (2023), Deformer Chen et al. (2022b), C2FViT Mok & Chung (2022), ModeT Wang et al. (2023), TM-DCA Chen et al. (2023), and XMorpher Shi et al. (2022). Their significantly larger receptive fields enable a more accurate understanding of spatial relationships between images. CorrMLP Meng et al. (2024) introduces the first MLP-based architecture for deformable medical image registration, overcoming transformers' limitations in capturing fine-grained long-range dependencies at full resolution due to high computational costs.

For each architecture, additional procedures can be integrated. Dual and inverse mechanisms are employed to ensure that two images deform symmetrically towards each other. Iterative and recursive methods are used to align images with significant displacements. Hierarchical, pyramid, and coarse-to-fine structures enhance the deformation field by leveraging high-resolution feature maps. Lastly, patch-wise techniques selectively sample a diverse range of features across a large search area while keeping computational overhead low.

## 2.2 CURRICULUM LEARNING (CL)

Imposing curriculum in neural networks can be traced back to Elman (1993). Inspired by the manners of how humans learn languages, Elman (1993) points out the importance of starting easily and gradually hardening the learning process when training networks. Bengio et al. (2009) further extends the idea to various vision and language tasks - where multi-stage curriculum strategies give rise to improved generalization and faster convergence. Named "curriculum learning" for these strategies, they are employed to different algorithms such as image classification Wang et al. (2019); Wei et al. (2021), object detection Zhang et al. (2019; 2017a), semantic segmentation Zhang et al. (2017b), self or semi supervised learning Murali et al. (2018), multi-task learning Sarafianos et al. (2017; 2018); Dong et al. (2017); Wang et al. (2018), multi-modal learning Gong et al. (2016); Gong (2017), etc Jiang et al. (2015); Matiisen et al. (2019); Weinshall et al. (2018). In the medical field, there has been limited prior research on curriculum learning (CL), with some examples in classification Jiménez-Sánchez et al. (2019); Luo et al. (2021), semantic segmentation Kervadec et al. (2019), self-supervised or semi-supervised learning Tang et al. (2018), and multi-modal learning Lotter et al. (2017). However, only a few studies leverage external knowledge from human experts, and even fewer integrate CL with image registration Burduja & Ionescu (2021); Zhou et al. (2023b). Our work introduces an automated CL approach for medical image registration that operates without the need of expert knowledge or prior experience.

## 3 PRELIMINARY: MEDICAL IMAGE REGISTRATION

Medical image registration, also known as image alignment, involves aligning two or more anatomically related images according to their spatial features. This process establishes non-linear dense correspondences between $n$-D medical images that are acquired from different patients, scanners, or at different times. It has been extensively studied due to its importance in clinical applications, such as monitoring tumor growth or conducting group analysis.

Deformable image registration is commonly framed as an optimization problem, aiming to minimize an energy function. This function generally consists of two main components: a penalty function that evaluates the similarity between the aligned and reference images, and a regularization term that enforces constraints on the registration field, such as promoting smoothness through a gradient loss penalty. In this process, the *fixed image* (or target image) serves as the baseline or template to which the *moving image* (or source image) is aligned. The fixed image provides the spatial coordinates used for alignment, while the moving image is adjusted to match the fixed image as closely as possible. The goal is to determine the optimal transformation that aligns the moving image with the fixed image. Below is a comprehensive explanation and formulation of a widely used framework for medical image registration.

The moving image $\hat{I_m}$ and the fixed image $I_f$ are initially transformed into a common coordinate system using an affine transformation. The affine-aligned moving image is referred to as $I_m$. Next, $I_m$ is deformed to align with $I_f$ using a deformation field $\phi$, which is generated by a specific network $f_\theta$ as

$$f_\theta(I_f, I_m) = \phi \qquad (1)$$

where $\theta$ is the parameters of the network. The overall loss function used for training the network is based on the energy function from traditional image registration techniques. This loss function comprises two components: one evaluates the similarity between the deformed image and the fixed image, while the other regularizes the deformation field to ensure smoothness. It is expressed as:

$$\mathcal{L}(I_f, I_m, \phi) = \mathcal{L}_{sim}(I_f, I_m, \phi) + \lambda \mathcal{R}(\phi), \qquad (2)$$

where $\mathcal{L}_{sim}$ denotes the image fidelity measure, and $\mathcal{R}$ is the regularization term for the deformation field. A common metric for evaluating the similarity between $I_f$ and $I_m$ is the local normalized cross-correlation:

$$LNCC(I_f, I_m, \phi) = \sum_{\mathbf{p} \in \Omega} \frac{\left( \sum_{\mathbf{p}_i} (I_f(\mathbf{p}_i) - \overline{I}_f(\mathbf{p}))([I_m \circ \phi](\mathbf{p}_i) - [\overline{I}_m \circ \phi](\mathbf{p})) \right)^2}{\left( \sum_{\mathbf{p}_i} (I_f(\mathbf{p}_i) - \overline{I}_f(\mathbf{p}))^2 \right) \left( \sum_{\mathbf{p}_i} ([I_m \circ \phi](\mathbf{p}_i) - [\overline{I}_m \circ \phi](\mathbf{p}))^2 \right)}, \qquad (3)$$

where $\overline{I}_f(\mathbf{p})$ and $\overline{I}_m(\mathbf{p})$ denote the mean value within the local window of size $n^3$ centered at voxel $\mathbf{p}$. A higher $LNCC$ indicates a better alignment, yielding the loss function: $\mathcal{L}_{sim}(I_f, I_m, \phi) = -LNCC(I_f, I_m, \phi)$. The regularizer $\mathcal{R}$ promotes similarity in displacement values between a given location and its neighboring locations. A commonly used diffusion regularizer can be formulated as:

$$\mathcal{R}_{diffusion}(\phi) = \sum_{\mathbf{p} \in \Omega} \| \nabla u(\mathbf{p}) \|^2, \qquad (4)$$

where $\nabla u$ is the spatial gradients of the displacement field $u$. The spatial gradients are approximated using forward differences, that is, $\frac{\partial u(\mathbf{p})}{\partial \{x,y,z\}} \approx u(\mathbf{p}_{\{x,y,z\}} + 1) - u(\mathbf{p}_{\{x,y,z\}})$.

## 4 METHOD

Research in deformable medical image registration has primarily focused on incremental performance improvements through minor changes to model architectures or the addition of new loss functions. While these adjustments can improve performance metrics, they often result in methods that are task- or dataset-specific, limiting their broader applicability. To overcome these limitations, we propose a novel approach that avoids modifying existing model architectures. Instead, we introduce a plug-in curriculum scheduler that integrates seamlessly with current methods. This scheduler, inspired by curriculum learning, organizes the learning process by increasing task difficulty based on two main criteria: sample difficulty and accuracy tolerance. For sample difficulty, we employ

two strategies: weighting challenging voxel-level samples within the brain and starting with blurred images before transitioning to sharper images as training progresses. The second criterion focuses on the rigor of supervision by utilizing ground truth data for more comprehensive training. This approach aims to improve the generalizability and flexibility of registration methods, making them more robust and widely applicable.

## 4.1 Sample Difficulty

Curriculum learning involves training models in a structured sequence, beginning with simpler examples and progressively advancing to more complex ones. A key challenge in this approach is developing automatic and objective metrics to assess the difficulty of each sample. However, in the medical domain, determining the difficulty of individual samples is not straightforward. For instance, identifying which specific MRI volume would be easier for learning the deformation field in registration tasks is not intuitive. Previous research has utilized domain knowledge from human experts to qualitatively assess the classification difficulty of medical images to guide curriculum learning. This approach, however, requires additional annotation efforts, depends on subjective human experience, and introduces potential bias. Instead of relying on this computationally intensive approach, we propose an automated method for assessing sample difficulties at two levels: voxel-level and volume-level.

### 4.1.1 Voxel-level sample difficulty

In medical image registration tasks, each voxel in a 3D volume contributes to the overall alignment between the moving and fixed images. In standard registration methods, all voxels within an MRI volume are assigned identical weights during training, based on the assumption that each voxel has equal significance. However, not all voxels are equally crucial for registration; for instance, areas like edges or regions with high contrast tend to be more difficult to align, while homogeneous areas are often easier. To address this, we propose a method to evaluate the varying difficulty in learning the displacement field between voxels. Based on this measure, we design a training schedule that gradually prioritizes voxels of different difficulty levels as training progresses.

**Difficulty Measure.** For a given MRI image $I$, it can be decomposed into a set of voxels $x_i$, where $i = \{1, ..., N\}$ and $N$ represents the total number of voxels in the image. The difficulty of these voxels is then evaluated using the Variance of Gradients (VoG) Agarwal et al. (2022) method. The VoG method is applied to the deformation fields in the following order.

The gradient calculation for each voxel involves determining how sensitive the deformation field is to changes in the input voxel values. Specifically, for each voxel in the input image, the gradient of the deformation field (displacement vector) is computed with respect to the voxel itself. This is expressed as $S_i = \frac{\partial D(x_i)}{\partial x_i}$, where $D(x_i)$ is the deformation vector at voxel $x_i$ and $S_i$ denotes the sensitivity of the deformation field to variations in that voxel. The gradient matrices for the deformation field are computed at different epochs or iterations during training, forming a series of checkpoints, denoted as $\{S_{i,1}, S_{i,2}, ..., S_{i,K}\}$, where $S_{i,t}$ represents the gradient matrix at checkpoint $t$. This process enables the monitoring of how the deformation field's sensitivity to voxel changes evolves over time. The average of the gradient matrices across all checkpoints for each voxel $x_i$ is then calculated as

$$\mu_i = \frac{1}{K} \sum_{t=1}^{K} S_{i,t}. \tag{5}$$

This mean represents the average sensitivity of the deformation field with respect to each voxel over time. Finally, the Variance of Gradients (VoG) is calculated across the checkpoints as:

$$VoG_i = \sqrt{\frac{1}{K} \sum_{t=1}^{K} (S_{i,t} - \mu)^2}. \tag{6}$$

This variance provides a measure of how much the deformation field's sensitivity to each voxel varies over time. A higher variance indicates that the voxel is more difficult to register, while a lower variance suggests that it is easier to register.

**Training Scheduler.** Once the difficulty of each voxel is determined through VoG, the loss function can be modified by multiplying it with calculated weights for each voxel: $L = \sum_i w_i \cdot \text{Loss}(x_i)$, adjusting the loss contribution during backpropagation.

To align with the goal of curriculum learning, which trains models by starting with simpler examples and gradually moving to more complex ones, the weights $w_i$ are scheduled to update at each epoch. This allows the model to adjust to new weights progressively as training proceeds. Each voxel $x_i$ is associated with an initial weight $w_{i,(1)}$ given by:

$$w_{i,(1)} = \frac{s_i}{\sum_{j=1}^{N} s_j} \tag{7}$$

where $s_i$ represents the rank of the voxel $x_i$ in the sorted VoG scores from highest to lowest, and $N$ is the total number of voxels in the MRI image. In this initial probability assignment, easier samples, which have lower VoG scores, are given higher probabilities. To reflect the changing importance of each voxel over training epochs, the weight for voxel $x_i$ at epoch $e$ is updated as:

$$w_{i,(e)} = w_{i,(e-1)} \times \lambda_i, \qquad \lambda_i = \sqrt[L]{\frac{1/N}{w_{i,(1)}}}. \tag{8}$$

Here, $L$ is the number of epochs over which the scheduling will last. By the final epoch, the probability for every voxel $x_i$ is set to:

$$w_{i,(final)} = 1/N. \tag{9}$$

This training scheduler ensures that the probability distribution over voxels smooths out over time, with each voxel eventually being assigned an equal probability by the end of the scheduling.

### 4.1.2 VOLUME-LEVEL SAMPLE DIFFICULTY

In the previous analysis, we identified variations in difficulty within a single MRI volume, leading to different contributions for individual voxels. On a broader scale, we aim to assess the difficulty across different volume samples.

When working with a set of MRI volumes, determining which volume poses a greater challenge in learning the deformation field is difficult. In situations where directly assessing the difficulty of data samples is not possible, a curriculum learning approach can be employed by deliberately simplifying the inputs. This is achieved by blurring the images with a Gaussian filter, reducing their complexity and information content Burduja & Ionescu (2021). Training begins with highly blurred images, gradually transitioning to sharper images as the process advances.

Specifically, at a given training epoch $e$, the degree of blur $\sigma$ is adjusted according to the following rule:

$$\sigma = \begin{cases} \sigma_{max} \times (1 - e/e_{sch}) & \text{if } e < e_{sch} \\ 0 & \text{if } e > e_{sch} \end{cases} \tag{10}$$

where $\sigma_{max}$ represents the initial level of blur at the start of training, and $e_{sch}$ is the final epoch during which the blur is applied.

### 4.2 ACCURACY TOLERANCE

In the context of model accuracy, curriculum learning generally involves presenting training data in a structured manner, starting from easier examples and gradually moving to more difficult ones to aid the model's learning process. An additional dimension to this approach is adjusting the level of strictness in supervision using ground truth data. This method is particularly effective in tasks that require mastering intricate details, such as medical imaging, where accurate alignment or segmentation is critical and develops over time.

In this approach, the level of strictness in evaluating the model against the ground truths is progressively increased. Early in training, the model is allowed more flexibility in its predictions (i.e., accepting a wider range of outputs as correct), focusing on general patterns. As the model progresses, the supervision becomes more stringent, requiring closer alignment with the exact ground truth (e.g., penalizing even slight deviations from the correct output). This strategy helps the model

avoid being overwhelmed by complex or highly detailed data in the early stages, enabling it to build a foundational understanding before being challenged to achieve higher accuracy levels.

At a given training epoch $e$ with a tolerance parameter $\epsilon_e$, the image similarity loss is defined as:

$$\mathcal{L}_\epsilon(\mathcal{L}_{sim}(I_f, I_m, \phi)) = \max\{\mathcal{L}_{sim}(I_f, I_m, \phi) + \epsilon, 0\} \tag{11}$$

when a higher similarity value corresponds to better alignment (e.g., for metrics like $LNCC$ or $SSIM$). The value of $\epsilon_e$ increases progressively as training continues. In contrast, when a lower similarity measure indicates better alignment (e.g., for metrics like $MSE$), the image similarity loss is given by:

$$\mathcal{L}_\epsilon(\mathcal{L}_{sim}(I_f, I_m, \phi)) = \max\{\mathcal{L}_{sim}(I_f, I_m, \phi) - \epsilon, 0\} \tag{12}$$

with the value of $\epsilon_e$ gradually decreasing as training progresses.

## 5 EXPERIMENTAL SETUP

### 5.1 DATASETS AND EVALUATION METRICS

Our method is employed in the analysis of two popular MRI databases, including IXI [1] and Mind-Boggle [2]. Detailed information regarding the datasets and their preparation procedures are presented in Section A.1 of the supplementary material. For quantitative comparisons on the datasets, Dice similarity coefficients (DSC) Dice (1945) and non-positive values in the Jacobian determinant (NJD) are computed. Specifically, DSC are measured between the segmentation labels of the fixed and moved images. The smoothness and invertibility of the predicted displacement fields are evaluated by determining the percentage of NJD of the deformation fields (i.e., % of $|J_\phi| \leq 0$). Adjusting the regularization parameter often involves a trade-off between registration accuracy and transformation smoothness Mok & Chung (2020b); Meng et al. (2022; 2024). Hence, registration methods should be evaluated using both DSC and NJD metrics. Further details and formulations can be found in Section A.2 of the supplementary material.

### 5.2 COMPARISON METHODS

Our method is extensively evaluated against state-of-the-art deformable image registration techniques, encompassing four traditional optimization-based methods (SyN Avants et al. (2008), NiftyReg Modat et al. (2010), LDDMM Beg et al. (2005), deedsBCV Heinrich et al. (2015)) and nine deep learning-based registration methods (VoxelMorph Balakrishnan et al. (2019), CycleMorph Kim et al. (2021), MIDIR Qiu et al. (2021), ViT-V-Net Chen et al. (2021), PVT Wang et al. (2021), CoTr Xie et al. (2021), nnFormer Zhou et al. (2021), TransMorph Chen et al. (2022a), H-ViT Ghahremani et al. (2024)). The methods and their hyperparameter settings are described in Section A.3 and A.4 of the supplementary material.

### 5.3 IMPLEMENTATION DETAILS

Our method builds upon two state-of-the-art deformable image registration models: TransMorph and H-ViT. These designs are referred to as TM+ and HV+, respectively. Both models were trained for 500 epochs on an NVIDIA A6000 GPU, using the Adam optimizer with a learning rate of $1 \times 10^{-4}$ and a batch size of 1. Hyperparameter selections are specified in the ablation study results. Section A.4 provides more details about the experiment settings.

## 6 RESULTS AND DISCUSSION

### 6.1 COMPARISON WITH BASELINE METHODS

We conducted inter-patient and atlas-to-patient registration experiments on both the IXI and Mind-Boggle datasets. Table 2 presents the registration metrics for IXI, comparing our method against several baselines using DSC and NJD. Our approach achieves the highest scores, including a DSC

---

[1] https://brain-development.org/ixi-dataset/
[2] https://mindboggle.info/

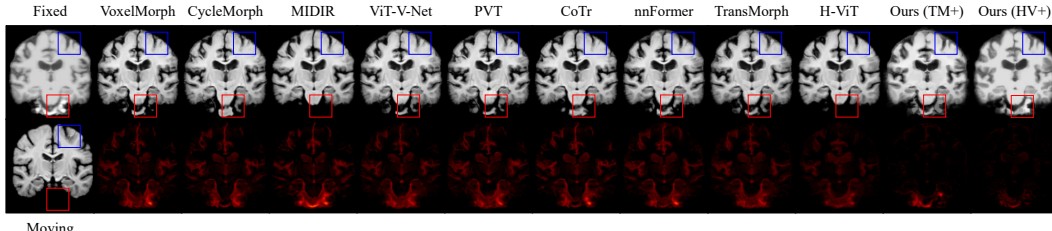

Moving

Figure 1: Example coronal slices from the IXI dataset and outcomes (moved MRI images and difference between fixed image and moved results) of baseline registration methods, compared with our methods. Red highlights indicate misalignments between the ground truth and the aligned results, while black highlights signify successful registration, with fewer red pixels indicating better performance.

| Method | Inter-patient Registration | | Atlas-to-patient Registration | |
|---|---|---|---|---|
| | DSC ↑ | NJD (%) ↓ | DSC ↑ | NJD (%) ↓ |
| SyN | $0.639 \pm 0.197$ | $< 0.001$ | $0.590 \pm 0.210$ | $< 0.001$ |
| NiftyReg | $0.626 \pm 0.183$ | $0.068 \pm 0.085$ | $0.587 \pm 0.223$ | $0.020 \pm 0.046$ |
| LDDMM | $0.730 \pm 0.134$ | $< 0.001$ | $0.638 \pm 0.186$ | $< 0.001$ |
| deedsBCV | $0.717 \pm 0.180$ | $0.188 \pm 0.059$ | $0.706 \pm 0.151$ | $0.147 \pm 0.050$ |
| VoxelMorph | $0.720 \pm 0.139$ | $0.799 \pm 0.103$ | $0.695 \pm 0.162$ | $1.586 \pm 0.339$ |
| CycleMorph | $0.704 \pm 0.167$ | $0.651 \pm 0.197$ | $0.706 \pm 0.155$ | $1.719 \pm 0.382$ |
| MIDIR | $0.721 \pm 0.156$ | $0.151 \pm 0.069$ | $0.711 \pm 0.158$ | $< 0.001$ |
| ViT-V-Net | $0.736 \pm 0.128$ | $0.999 \pm 0.201$ | $0.702 \pm 0.155$ | $1.609 \pm 0.319$ |
| PVT | $0.733 \pm 0.117$ | $1.314 \pm 0.600$ | $0.695 \pm 0.159$ | $1.858 \pm 0.314$ |
| CoTr | $0.741 \pm 0.132$ | $0.719 \pm 0.269$ | $0.706 \pm 0.164$ | $1.298 \pm 0.343$ |
| nnFormer | $0.744 \pm 0.130$ | $0.800 \pm 0.283$ | $0.719 \pm 0.157$ | $1.595 \pm 0.358$ |
| TransMorph | $0.763 \pm 0.119$ | $0.617 \pm 0.210$ | $0.724 \pm 0.150$ | $1.502 \pm 0.342$ |
| H-ViT | $0.779 \pm 0.078$ | $0.589 \pm 0.182$ | $0.740 \pm 0.139$ | $0.707 \pm 0.185$ |
| Ours (TM+) | $0.808 \pm 0.063$ | $0.493 \pm 0.101$ | $0.755 \pm 0.107$ | $0.401 \pm 0.090$ |
| Ours (HV+) | $0.812 \pm 0.060$ | $0.285 \pm 0.071$ | $0.772 \pm 0.091$ | $< 0.001$ |

Table 2: Quantitative evaluation results for the registration methods on the IXI dataset for 34 anatomical structures over 115 random pairs for inter-patient and 115 pairs for atlas-to-patient registrations.

performance improvement of $+0.045$ and $+0.033$ in inter-patient registration compared to TransMorph and H-ViT, respectively. While the differences between the baselines are relatively small, our method shows a significantly larger margin, yielding meaningful results. Figure 1 presents the visualized registration results, highlighting the differences between the fixed image and the transformed outputs as well. Our method shows more precise warping of the moving MRI compared to other approaches, especially in the areas marked by the red and blue rectangles. The registration results for the MindBoggle dataset are shown in Table 3, where our method achieves an increase in DSC values by $+0.056$ and $+0.070$ compared to TransMorph and H-ViT, respectively, in atlas-to-patient registration experiments, representing a substantial performance improvement. Detailed results, including DSC values for each anatomical structure, are provided in the Supplementary material in Section B.

## 6.2 VISUALIZATION RESULTS BY EPOCHS

In Figure 2, the progression of the moved image at key training epochs is illustrated. Our curriculum learning-based approach initially focuses on learning coarse spatial transformations during the early stages of training, deliberately omitting intricate details to prioritize broader structural changes. By starting with simpler, larger-scale adjustments, the model establishes a solid foundation for subsequent refinement. As training progresses, the model gradually hones in on finer, localized regions, refining these areas based on the broader shapes learned in the earlier stages. This phased approach allows the network to incrementally build upon its initial understanding, ultimately leading to more accurate and precise registration results. By progressively tackling the complexity of the transformation, this training strategy significantly enhances the overall performance of the registration process, resulting in improved alignment and greater detail capture in the final output.

| Method | Inter-patient Registration | | Atlas-to-patient Registration | |
|---|---|---|---|---|
| | DSC ↑ | NJD (%) ↓ | DSC ↑ | NJD (%) ↓ |
| VoxelMorph | $0.674 \pm 0.197$ | $0.821 \pm 0.170$ | $0.666 \pm 0.201$ | $0.831 \pm 0.163$ |
| CycleMorph | $0.679 \pm 0.194$ | $1.044 \pm 0.211$ | $0.671 \pm 0.199$ | $1.064 \pm 0.189$ |
| MIDIR | $0.637 \pm 0.197$ | $0.403 \pm 0.215$ | $0.539 \pm 0.292$ | $0.347 \pm 0.205$ |
| ViT-V-Net | $0.700 \pm 0.186$ | $1.168 \pm 0.225$ | $0.695 \pm 0.187$ | $0.840 \pm 0.573$ |
| PVT | $0.588 \pm 0.214$ | $2.006 \pm 0.254$ | $0.583 \pm 0.216$ | $2.034 \pm 0.217$ |
| CoTr | $0.633 \pm 0.214$ | $0.691 \pm 0.163$ | $0.630 \pm 0.218$ | $0.701 \pm 0.141$ |
| nnFormer | $0.622 \pm 0.210$ | $1.077 \pm 0.210$ | $0.618 \pm 0.213$ | $1.090 \pm 0.189$ |
| TransMorph | $0.699 \pm 0.186$ | $0.702 \pm 0.106$ | $0.695 \pm 0.189$ | $0.716 \pm 0.082$ |
| H-ViT | $0.731 \pm 0.170$ | $0.328 \pm 0.061$ | $0.726 \pm 0.173$ | $0.335 \pm 0.049$ |
| Ours (TM+) | $0.749 \pm 0.126$ | $0.209 \pm 0.039$ | $0.751 \pm 0.086$ | $0.199 \pm 0.020$ |
| Ours (HV+) | $0.781 \pm 0.099$ | $< 0.001$ | $0.796 \pm 0.062$ | $0.128 \pm 0.011$ |

Table 3: Quantitative evaluation results for the registration methods on the MindBoggle dataset for 41 anatomical structures over 114 random pairs for inter-patient and 222 pairs for atlas-to-patient registrations.

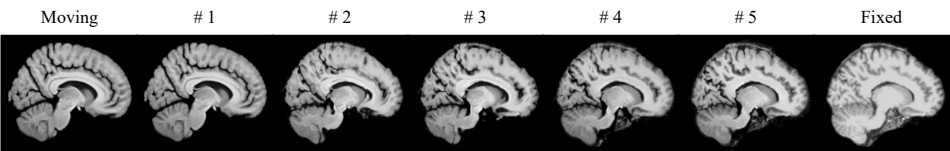

Figure 2: The visualization of moved MRI images at the end of each training stage.

## 6.3 ABLATION STUDY

### 6.3.1 COMPONENTS OF CURRICULUM LEARNING

We perform an ablation study on the components of curriculum learning: voxel-level sample difficulty (VLSD), volume-level sample difficulty (VMSD), and accuracy tolerance (AT), with the resulting metrics shown in Table 4. Incorporating all three components in the learning schedule yields the best performance.

| Components | VLSD | VMSD | AT | VLSD, VMSD | VLSD, AT | VMSD, AT | Full |
|---|---|---|---|---|---|---|---|
| DSC ↑ | 0.754 ±0.110 | 0.745 ±0.132 | 0.749 ±0.124 | 0.761 ±0.105 | 0.767 ±0.099 | 0.758 ±0.127 | **0.772 ±0.091** |

Table 4: Ablation study on the curriculum learning components for the IXI registration.

### 6.3.2 PARAMETERS OF VOXEL-LEVEL SAMPLE DIFFICULTY

**Sample difficulties.** In the curriculum for voxel-level sample difficulty, we update the weights of voxels at each epoch to encourage the network to increasingly focus on challenging samples. We contrast this with a simpler approach that maintains a fixed difficulty level for each voxel, utilizing only the difficulty measure and omitting the training scheduler. Table 5 presents the DSC measures for various fixed difficulty values. Our method demonstrates superior performance by progressively increasing the probabilities for difficult samples, allowing the network to concentrate more on challenging details.

**Scheduling epochs.** An ablation study is conducted on different scheduling epochs for voxel-level sample difficulty. Here, $e_{start}$ indicates the epoch at which scheduling begins; thus, the VoG values for individual voxels are computed and stacked for the difficulty measure during the epochs prior to $e_{start}$. Meanwhile, $e_{end}$ signifies the epoch when scheduling concludes. Consequently, the difficulty weights of voxels are updated between $e_{start}$ and $e_{end}$, after which equal probabilities are assigned to all voxels. Note that $e_{start} = K$ and $e_{end} = K + L$, with $K$ and $L$ detailed in Section 4.1.1. Table 6 shows the DSC score of methods trained with varying scheduling epochs, showing that setting $e_{start}$ to 100 epochs and $e_{end}$ to 300 epochs yields the best DSC measures.

| $w_i$ | $w_{i,(1)}$ | $w_{i,(final/4)}$ | $w_{i,(final/2)}$ | $w_{i,(final)}$ | Ours |
|---|---|---|---|---|---|
| DSC ↑ | $0.750 \pm 0.135$ | $0.767 \pm 0.105$ | $0.764 \pm 0.119$ | $0.758 \pm 0.127$ | $\mathbf{0.772 \pm 0.091}$ |

Table 5: Ablation study on methods with fixed difficulties of voxel-level sample difficulty for the IXI registration.

| $e_{start}, e_{end}$ | 100, 200 | 100, 300 | 200, 300 | 200, 400 |
|---|---|---|---|---|
| DSC ↑ | $0.765 \pm 0.097$ | $\mathbf{0.772 \pm 0.091}$ | $0.767 \pm 0.102$ | $0.760 \pm 0.099$ |

Table 6: Ablation study on the scheduling epochs of voxel-level sample difficulty for the IXI registration.

### 6.3.3 PARAMETERS OF VOLUME-LEVEL SAMPLE DIFFICULTY

The effects of varying degrees of initial blur and scheduling epochs are presented in Table 7. The ablation study on initial blur was conducted using a fixed scheduling epoch of $e_{sch} = 300$, while the ablation study on scheduling epochs was carried out with a fixed initial blur value of $\sigma_{max} = 1.0$. Our optimal method is configured with hyperparameters of $e_{sch} = 300$ and $\sigma_{max} = 1.0$.

| $\sigma_{max}$ | 0.5 | 0.75 | 1.0 | 1.5 | 2.0 |
|---|---|---|---|---|---|
| DSC ↑ | $0.760 \pm 0.119$ | $0.769 \pm 0.098$ | $\mathbf{0.772 \pm 0.091}$ | $0.767 \pm 0.101$ | $0.757 \pm 0.125$ |
| $e_{sch}$ | 200 | 250 | 300 | 350 | 400 |
| DSC ↑ | $0.765 \pm 0.120$ | $0.768 \pm 0.115$ | $\mathbf{0.772 \pm 0.091}$ | $0.763 \pm 0.116$ | $0.759 \pm 0.120$ |

Table 7: Ablation study on the initial blur and the scheduling epoch of volume-level sample difficulty for the IXI registration.

### 6.3.4 PARAMETERS OF ACCURACY TOLERANCE

The DSC measurements for various methods trained with different $\epsilon$ values are presented in Table 8. The chosen $\epsilon$ values were determined by analyzing the model's training curve without any tolerances. Our optimal method is configured with the following $\epsilon$ values: 0.24, 0.26, 0.28, 0.3, 0.31, 0.32, 0.34, and 0.38 for epochs under 3, 5, 30, 100, 200, 300, 400, and 500, respectively. It is important to note that these values increase as training progresses, as we utilize the image similarity metric $LNCC$.

| | $\epsilon$ (epochs) | DSC ↑ |
|---|---|---|
| # 1 | $0.24(< 3), 0.26(< 5), 0.28(< 30), 0.3(< 100),$ $0.31(< 200), 0.32(< 300), 0.34(< 400), 0.38(< 500)$ | $\mathbf{0.772 \pm 0.091}$ |
| # 2 | $0.24(< 5), 0.26(< 30), 0.29(< 100), 0.3(< 150),$ $0.305(< 200), 0.31(< 300), 0.45(< 500)$ | $0.754 \pm 0.141$ |
| # 3 | $0.24(< 5), 0.26(< 30), 0.29(< 100), 0.3(< 150),$ $0.305(< 200), 0.31(< 270), 0.32(< 400), 0.5(< 500)$ | $0.748 \pm 0.160$ |

Table 8: Ablation study on the accuracy tolerance scheduling for the IXI registration.

## 7 CONCLUSION

We introduced an innovative approach that marks a notable advancement in deformable medical image registration through the development of a plug-in scheduler inspired by curriculum learning. This technique improves the adaptability and flexibility of existing network architectures without the need for substantial alterations, enabling broader applicability across diverse tasks and datasets. By concentrating on the dual aspects of sample difficulty and matching accuracy, we effectively steer the network's learning process towards achieving accurate image alignment. Our distinctive application of the Variance of Gradients (VoG) for assessing voxel difficulty and the gradual shift from blurred to clearer images for volume difficulty sets our work apart from conventional methods. In contrast to earlier curriculum learning techniques in medical imaging, our automated solution removes the necessity for expert knowledge, making it accessible to a wider audience.

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

SUPPLEMENTARY MATERIAL

# A  EXPERIMENTAL SETUP

## A.1  DATASETS AND PREPROCESSING

**IXI (Information eXtraction from Images).** The publicly available IXI [3] dataset contains 576 T1-weighted MRI scans. These were split into 403 for training, 58 for validation, and 115 for testing. The MRI volumes were cropped to dimensions of $160 \times 192 \times 224$ and underwent preprocessing using FreeSurfer. Registration performance was assessed using label maps corresponding to 34 anatomical structures. For inter-patient registration inference, 115 pairs were randomly selected for the primary evaluation of the methods. In the atlas-to-patient registration tasks, the IXI images served as the fixed images, while the moving image was an atlas brain MRI from Kim et al. (2021).

**MindBoggle-101.** The MindBoggle dataset [4] contains 41 anatomically labeled brain surfaces from 101 healthy individuals, divided into four subsets: HLN (12 scans), MMRR (23 scans), NKI (42 scans), and OASIS (20 scans). MRI volumes from the HLN, MMRR, and NKI subsets were registered to MNI152 space [5] using affine transformations, with a resolution of $1 \times 1 \times 1mm^3$ and a voxel grid size of $160 \times 192 \times 224$. Registration performance was evaluated based on label maps of 41 anatomical structures. For inter-patient registration, 15, 33, and 66 pairs were randomly selected from the HLN, MMRR, and NKI subsets, respectively. In the patient-to-atlas registration task, one random sample from each subset was chosen as the atlas, and the remaining samples were registered to it, with the process repeated twice to produce 33, 66, and 123 registration pairs for the HLN, MMRR, and NKI subsets, respectively.

## A.2  EVALUATION METRICS

**Dice Score.** The Dice score Dice (1945), also known as the Dice coefficient, is a statistical measure used to gauge the similarity between two sets, commonly applied in image segmentation. It quantifies the overlap between the predicted segmentation and the ground truth by calculating the ratio of twice the area of overlap to the total number of pixels in both sets. The formula is:

$$\text{Dice} = \frac{2 \times |A \cap B|}{|A| + |B|} \tag{13}$$

where $|A|$ and $|B|$ are the sizes of the two sets (predicted and ground truth), and $|A \cap B|$ is the size of their intersection. The Dice score ranges from 0 to 1, with 1 indicating perfect agreement and 0 indicating no overlap.

**Jacobian determinant.** The Jacobian determinant measures local volume changes induced by a deformation field. It is computed from the Jacobian matrix, which contains partial derivatives of the deformation field with respect to spatial coordinates. A positive Jacobian determinant indicates a local volume expansion or contraction, while a non-positive Jacobian determinant indicates a problematic deformation, such as folding or inversion of the image. The percentage of non-positive values in the Jacobian determinant provides an indication of the quality of the deformation; ideally, this percentage should be low to ensure topological preservation and avoid unrealistic transformations.

## A.3  COMPARISON METHODS

Four traditional optimization-based methods and nine deep learning-based registration methods are introduced as baselines for the IXI atlas-to-patient registration task, as outlined below.

**SyN** Avants et al. (2008) introduces a novel symmetric image normalization method to maximize cross-correlation within the space of diffeomorphic maps, along with the necessary Euler-Lagrange equations for optimization.

---

[3]`https://brain-development.org/ixi-dataset/`
[4]`https://mindboggle.info/`
[5]`https://www.lead-dbs.org/about-the-mni-spaces/`

**NiftyReg** Modat et al. (2010) presents a GPU-optimized, parallel-friendly algorithm that performs MR image registration in less than one minute, achieving the same accuracy as conventional serial methods for segmentation propagation.

**LDDMM** Beg et al. (2005) explores the Euler-Lagrange equations for large deformation diffeomorphic metric mapping, deriving the minimizing vector fields and implementing a semi-Lagrangian method to compute particle flows and metric distances on anatomical structures.

**deedsBCV** Heinrich et al. (2015) presents a automated discrete medical image registration framework for multi-organ segmentation across various modalities, using local self-similarity context (SSC) for similarity assessment and a Markov random field (MRF) to ensure smoothness efficiently.

**VoxelMorph** Balakrishnan et al. (2019) formulates registration as a function mapping an input image pair to a deformation field aligned by a CNN, using two training strategies: an unsupervised method maximizing image intensity matching and a second method leveraging auxiliary segmentations from the training data.

**CycleMorph** Kim et al. (2021) proposes a cycle-consistent deformable image registration method that enhances performance by preserving topology during deformation, applicable to both 2D and 3D problems and easily extendable to multi-scale implementations for large volume registration.

**MIDIR** Qiu et al. (2021) introduces a deep learning registration framework for fast mono-modal and multi-modal image registration using differentiable mutual information and B-spline free-form deformation to achieve smooth, efficient diffeomorphic deformation.

**ViT-V-Net** Chen et al. (2021) integrates ViT and ConvNets to improve volumetric medical image registration, drawing inspiration from ViT-based image segmentation methods that combine ConvNets for better localization.

**PVT** Wang et al. (2021) addresses dense prediction tasks by providing high output resolution and lower computational costs than ViT, while combining the strengths of both CNNs and transformers as a versatile backbone for various vision applications.

**CoTr** Xie et al. (2021) proposes a framework that combines CNNs with a deformable Transformer (DeTrans) for accurate 3D medical image segmentation, efficiently addressing long-range dependencies while reducing computational complexities by focusing on key positions through deformable self-attention.

**nnFormer** Zhou et al. (2021) is a 3D transformer for volumetric medical image segmentation that integrates interleaved convolution with self-attention, employs local and global volume-based self-attention mechanisms, and enhances the U-Net architecture by replacing skip connections with skip attentions.

**TransMorph** Chen et al. (2022a) is a hybrid model combining Transformer and ConvNet architectures for volumetric medical image registration, featuring diffeomorphic variants that ensure topology preservation and a Bayesian variant for assessing registration uncertainty.

**H-ViT** Ghahremani et al. (2024) introduces a deformable image registration method that uses dual self-attention and cross-attention mechanisms to capture multi-scale flow features, enabling high-level features to inform the representation of low-level ones across spatially distant voxel patches.

## A.4 IMPLEMENTATION DETAILS

All registration models, including the baselines and our proposed method, were trained for 500 epochs on an NVIDIA A6000 GPU, using the Adam optimizer with a learning rate of $1 \times 10^{-4}$ and a batch size of 1. For the competing methods, the default network parameter settings recommended by their respective authors were applied.

**SyN** Avants et al. (2008): For both inter-patient and atlas-to-patient brain MRI registration tasks, the mean squared difference (MSQ) is employed as the objective function, applying a default Gaussian smoothing of 3 and utilizing three scales with 180, 80, and 40 iterations, respectively.

**NiftyReg** Modat et al. (2010): The sum of squared differences (SSD) is employed as the objective function, while bending energy acts as a regularizer for all registration tasks. In inter-patient brain MRI registration, the regularization weight is empirically set to 0.0002, utilizing three scales with

300 iterations each. For atlas-to-patient brain MRI registration, the regularization weight is modified to 0.0006, with three scales and 500 iterations applied for each scale.

**LDDMM** Beg et al. (2005): The mean squared error (MSE) serves as the default objective function. For both inter-patient and atlas-to-patient brain MRI registration, a smoothing kernel size of 5, a smoothing kernel power of 2, a matching term coefficient of 4, a regularization term coefficient of 10, and an iteration count of 500 is applied.

**deedsBCV** Heinrich et al. (2015): The default objective function is self-similarity context (SSC). For both inter-patient and atlas-to-patient brain MRI registration, the hyperparameter values recommended by Hoffmann et al. (2020) is utilized for neuroimaging, setting the grid spacing, search radius, and quantization step to $6 \times 5 \times 4 \times 3 \times 2$, $6 \times 5 \times 4 \times 3 \times 2$, and $5 \times 4 \times 3 \times 2 \times 1$, respectively.

**VoxelMorph** Balakrishnan et al. (2019): For inter-patient and atlas-to-patient brain MRI registration, the regularization hyperparameter $\lambda$ is set to 0.02 and 1, respectively, as these values are identified as optimal by the authors.

**CycleMorph** Kim et al. (2021): In CycleMorph, the hyperparameters $\alpha$, $\beta$, and $\lambda$ denote the weights for cycle loss, identity loss, and deformation field regularization, respectively. For inter-patient brain MRI registration, the hyperparameters are set to $\alpha = 0.1$, $\beta = 0.5$, and $\lambda = 0.02$, whereas for atlas-to-patient brain MRI registration, they are set to $\alpha = 0.1$, $\beta = 0.5$, and $\lambda = 1$. The authors suggest these values as optimal for neuroimaging.

**MIDIR** Qiu et al. (2021): The same loss function and $\lambda$ value as those used in VoxelMorph are applied. Additionally, the control point spacing $\delta$ for the B-spline transformation was set to 2 for all tasks, which was identified as the optimal value by the authors.

**ViT-V-Net** Chen et al. (2021): This registration network was built on the Vision Transformer (ViT) framework Dosovitskiy et al. (2021). The default network hyperparameter settings recommended by the authors are utilized.

**PVT** Wang et al. (2021): In the context of registration with the PVT model, we adhered to the configuration recommended by TransMorph. Specifically, the default settings are implemented, with the exception that the embedding dimensions are adjusted to $\{20, 40, 200, 320\}$, the number of heads is set to $\{2, 4, 8, 16\}$, and the depth is increased to $\{3, 10, 60, 3\}$ to ensure a comparable number of parameters to those in TransMorph.

**CoTr** Xie et al. (2021): Default network settings by the authors are used for all registration tasks.

**nnFormer** Zhou et al. (2021): To ensure a fair comparison, the same Transformer parameter values from TransMorph are used for nnFormer, as nnFormer is also built on the Swin Transformer Liu et al. (2021) architecture.

**TransMorph** Chen et al. (2022a): The same loss function parameters as those used in VoxelMorph are applied to all tasks of TransMorph.

**H-ViT** Ghahremani et al. (2024): The default network settings provided by the authors are applied to all registration tasks. Specifically, the encoder consists of five layers with sizes $[32, 64, 128, 256, 512]$ and the decoder with sizes $[192, 192, 192, 192]$. The parameters for H-ViT are configured as follows: the embedding dimension $f_e$ is set to 192, the number of feature maps $S_h$ to 4, the voxel patch size to $2 \times 2 \times 2$, the model depth to 1, the MLP ratio in the feedforward network to 2, the drop rate to 0, and the number of heads to 32.

# B  RESULTS AND DISCUSSION

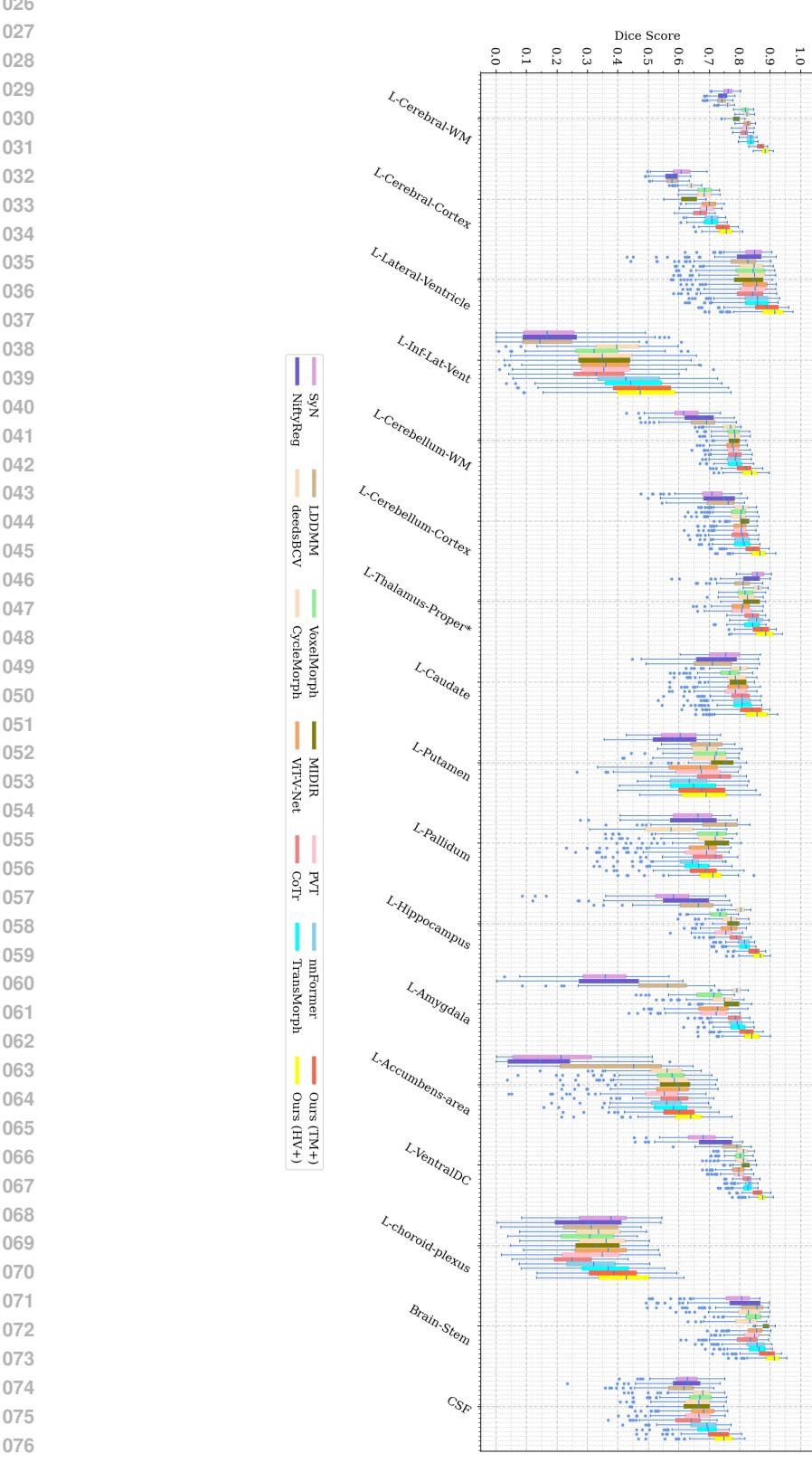

Figure 3: Dice score results for the atlas-to-patient registration of various methods on the IXI dataset (continued).

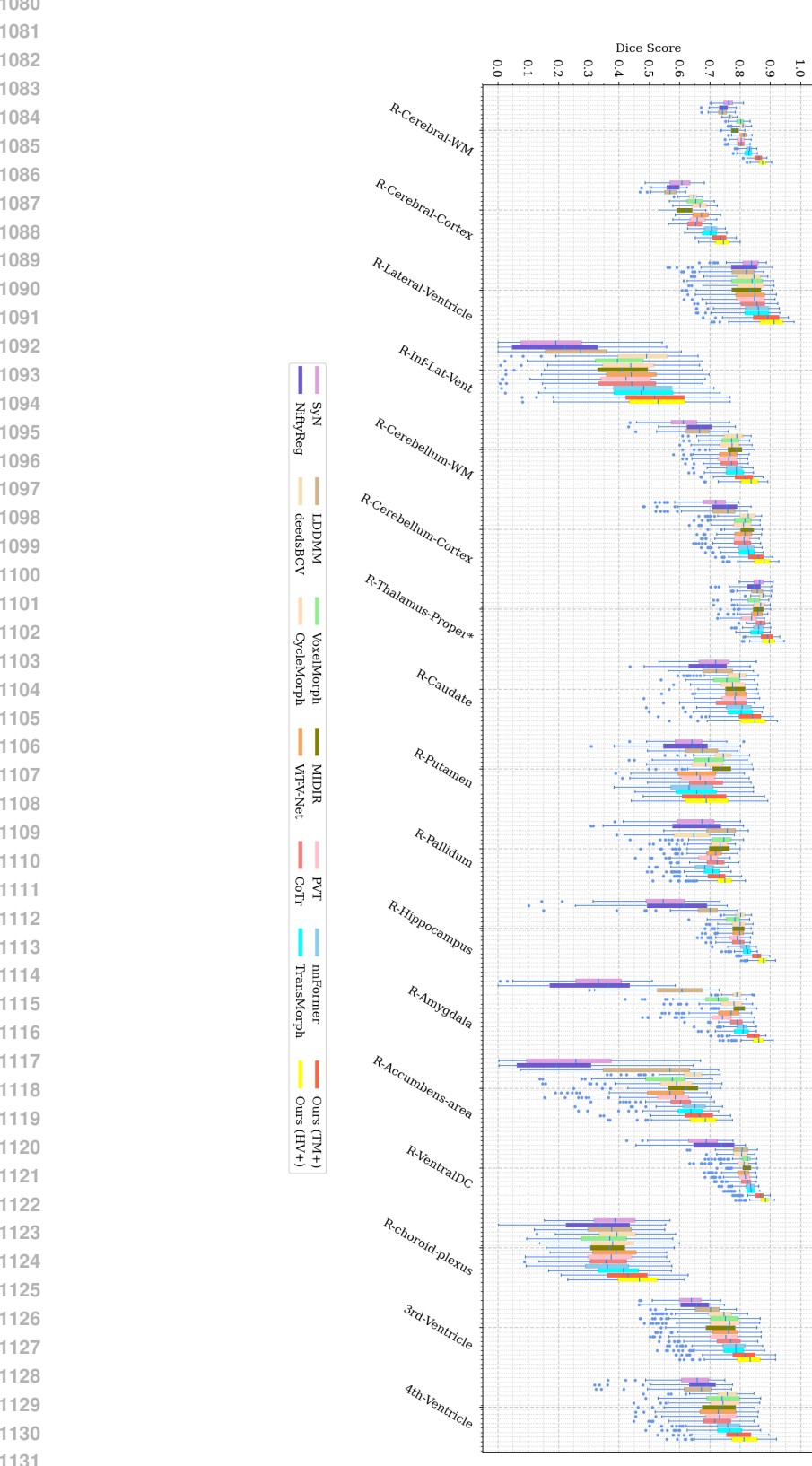

Figure 4: Dice score results for the atlas-to-patient registration of various methods on the IXI dataset.

