# OpenReview forum: "A Plug-In Curriculum Scheduler for Improved Deformable Medical Image Registration"
_ICLR.cc/2025/Conference — ICLR 2025 Conference Withdrawn Submission_

### Official Review · Reviewer_to2a · 2024-10-20

**Soundness:** 2
**Presentation:** 2
**Contribution:** 3
**Rating:** 6
**Confidence:** 4

**Summary:**

The paper proposed a new scheduler which could directly be plug into current networks. This curriculum learning-based scheduler offers automated curriculum learning solution on medical imaging registration without having prior knowledge. The scheduler has been tested on brain registrations of two different dataset.

**Strengths:**

The curriculum learning-based scheduler allows the users to directly plug in the existing models without any other modification, which makes the work appealing. Although the “Difficulty Measure” and the “Training Scheduler” parts are similar to a cited paper (Zhou et al., 2023b), this paper certainly turned the former method that applied the image classification to the current one that fits image registration, and it also made some modification on the blurring adjustment and the accuracy tolerance. The paper is also clear to read.

**Weaknesses:**

1. The paper has compared the registration on two datasets. However, both are inter-patient and atlas-to-patient brain registration, which might undermine the generality of the conclusion that the proposed scheduler could increase registration accuracy.
2. The two learning-based models the paper added the scheduler to (i.e. TransMorph & H-ViT) don’t use any scheduler in the origin (constant learning rate). Adding other schedulers might also increase the performance.

**Questions:**

1. Why using learning-based network that designed for image segmentation in the experiments? (CoTr and nnFormer).
2. It would be better to change one dataset to non-brain registration (e.g. NLST lung registration dataset available in learn2reg)
3. It might be more convincing to test other common schedulers (e.g. cosine scheduler) on the TransMorph & H-ViT to verify that those won’t further increase the registration performance but the proposed one can.

---

### Official Review · Reviewer_9jqP · 2024-11-01

**Soundness:** 3
**Presentation:** 3
**Contribution:** 2
**Rating:** 3
**Confidence:** 5

**Summary:**

This paper proposes a plug-in curriculum scheduler that can be integrated into existing deep learning registration methods, which extends the concept of curriculum learning to medical image registration and enables improved performance on brain MRI registration tasks.

**Strengths:**

This paper was well-written and easy to follow. It provides a comprehensive summary of existing literature in the field of medical image registration and identifies the key obstacle of existing methods, i.e., (i) incremental performance gains by adjusting network architectures or introducing new loss functions, and (ii) being tailored to specific tasks or datasets with limited general applicability.

**Weaknesses:**

Despite that the key obstacle of existing methods were successfully identified, this paper still did not offer a satisfactory solution:
Firstly, the improvements gained from the proposed method are also incremental by adding a new training scheduler. This paper extends the concept of curriculum learning to medical image registration, which shows limited technical contribution as the technical modification made for registration tasks is not significant: the voxel-wise and volume-wise difficulty measures are not new (Variance of Gradients and Gaussian filter). The introduction of curriculum learning into medical image registration is also not radically novel as this has been explored for registration in Burduja & Ionescu (2021)’s study. Section 2.2 mentioned Burduja & Ionescu (2021)’s study but failed to clearly discuss the differences.
Secondly, the proposed method is claimed to be generalizable across diverse network architectures, tasks, and datasets. However, the experiments were limited to two exsiting registration models (TransMorph, H-ViT) and two brain MRI datasets (IXI, Mindboggle), not demonstrating its generalizability. Provided that there are many well-benchmarked datasets for medical registration, the experiments should includes more evaluation tasks such as the registration of cardiac MRI or lung CT. The proposed curriculum scheduler also should be embedded with more existing registration methods for evaluation. In addition, according to the ablation studies, the performance is sensitive to the hyperparameters. This reduces the generalizability as hyperparameters might be re-chosen for different datasets and tasks.

**Questions:**

1. How was the proposed method compared to the CorrMLP and IIRP-Net? These two methods were mentioned in the introduction but missing in the comparison.
2. The reported results on the Mindboggle datasets are much higher than those reported by most of existing literature. This is possibly because only 41 anatomical structures were used for calculating DSC. Why? The Mindboggle datasets provided labels for 62 structures.
3. It is unclear that the ablation studies were performed on which dataset and with which registration model?

---

### Official Review · Reviewer_ZgVA · 2024-11-04

**Soundness:** 3
**Presentation:** 3
**Contribution:** 3
**Rating:** 3
**Confidence:** 5

**Summary:**

This paper introduces a plug-in curriculum scheduler for medical image registration that can be integrated into existing registration networks without altering the architecture or requiring expert knowledge. The proposed method measures sample difficulty using gradient variance and progressively schedules training by re-weighting the loss for each voxel during backpropagation. Experiments on various brain image registration tasks, including inter-patient and atlas-to-patient registration, demonstrate the effectiveness of this approach.

**Strengths:**

1. The authors provide valuable insight by recognizing that not all voxels are equally important for image registration. They propose using gradient variance during backpropagation to adjust voxel weights, introducing a novel training scheme to the image registration community.

2. The curriculum training approach is also commendable for its simplicity and ease of integration into existing registration frameworks, requiring minimal additional effort.

**Weaknesses:**

1. In Lines 240-241, the authors state, “areas like edges or regions with high contrast tend to be more difficult to align, while homogeneous areas are often easier.” This contradicts findings in foundational optical flow research [1] and the aperture problem, which suggest otherwise.

2. While the authors argue that not all voxels are equally important for registration, there is no visual illustration of the gradient variance for these voxels to substantiate this claim.

3. The paper claims improved performance across various tasks and datasets, yet only includes brain image registration experiments, which involve small and relatively simple deformations.

4. It is unclear how assigning different weights to each voxel enhances registration performance, as this does not appear to address issues like the aperture problem, large deformations, or multi-modal registration challenges based on the description provided.

5. The implementation details for baseline methods are unclear, making it difficult to assess the fairness of comparisons. To ensure robust evaluation, the authors are encouraged to submit their results to an online leaderboard, such as the Learn2Reg Challenge, which addresses unsupervised brain image registration [2].

[1] Horn, Berthold KP, and Brian G. Schunck. "Determining optical flow." Artificial intelligence 17.1-3 (1981): 185-203.

[2] https://learn2reg.grand-challenge.org/evaluation/l2r24-lumir/leaderboard/

**Questions:**

1. Can the authors provide a gradient variance heatmap of a specific sample throughout the training process?

2. Regarding Weakness Point 1, can the authors clarify their statement about edges and high-contrast regions being harder to align? Additionally, could they connect this explanation with the gradient variance heatmap to reinforce their claims?

3. In Table 2 of the H-ViT paper [1], the reported Dice scores on the IXI dataset are significantly higher than those presented here: 0.810 vs. 0.779 for inter-patient registration and 0.797 vs. 0.740 for atlas-to-patient. Could the authors explain this discrepancy?

[1] Ghahremani, Morteza, et al. "H-ViT: A Hierarchical Vision Transformer for Deformable Image Registration." Proceedings of the IEEE/CVF Conference on Computer Vision and Pattern Recognition. 2024.

---

### Official Review · Reviewer_eEyW · 2024-11-04

**Soundness:** 3
**Presentation:** 3
**Contribution:** 3
**Rating:** 5
**Confidence:** 5

**Summary:**

This work explores the improvement of image registration task on the direction of training process. Instead of adjusting the network structure or introducing new loss function, the authors proposed a plug-in curriculum scheduler to schedule the training starting from easier samples to more difficult samples with higher tolerance to more strict constraints. The experiments on IXI and Mindboogle datasets show the improvement from the plug-in scheduler.

**Strengths:**

1. This work provides an simple but effective plug-in module that could benefit many existing image registration models by utilizing curriculum learning.
2. Three strategies are proposed to guide the model training, voxel-level sample difficulty (VLSD), volume-level sample difficulty (VMSD), and accuracy tolerance (AT).
3. The writing is good and easy to follow.

**Weaknesses:**

There are some experiments I'd like to evaluate the benefit brought by this "universal" plug-in module.
1. I'm interested in seeing how the curriculum learning approach affects convergence speed or final performance at different training durations, therefore I'd like to see the improvement with different training iteration settings, eg 100, 200, 500, 1000, etc.
2. Before this work, there are some "plug-in module" works exploring the improvement of registration like [1][2]. I'd like to compare the improvement with these works in terms of performance gains, computational overhead. And very curious to see the improvement combining this work and [1][2] as they provide the different designs in terms of training procedure.
3. As this is a universal plug-in module, the experiments should also be conducted in other regions, for example, thorax or abdomen, etc.
4. Overall, the novelty of this work is more application and engineering oriented. The novelty in this work is how to design the curriculum learning for image registration task.


[1] Unsupervised 3d registration through optimization-guided cyclical self-training. MICCAI 2023
[2] On-the-Fly Guidance Training for Medical Image Registration. MICCAI 2024

**Questions:**

More experiments are necessary to demonstrate the improvement and benefit of curriculum learning.

---

### Note · Authors · 2025-01-08

I have read and agree with the venue's withdrawal policy on behalf of myself and my co-authors.